# Current definitions of advanced multimorbidity: a protocol for a scoping review

Sarah P Bowers ,[1] Polly Black,[1] Lewis McCheyne,[2] Darcy Wilson,[3]
Sarah E E Mills,[1,2] Utkarsh Agrawal,[4] Linda Williams,[5] Frances Quirk,[1,2]
Jo Bowden[1,2]

[1]University of St Andrews School of Medicine, St Andrews, UK
[2]NHS Fife, Kirkcaldy, UK
[3]NHS Tayside, Dundee, UK
[4]University of Oxford Nuffield Department of Primary Care Health Sciences, Oxford, UK
[5]Usher Institute, University of Edinburgh, Edinburgh, UK

**Correspondence to**
Dr Sarah P Bowers;
sb461@st-andrews.ac.uk

## ABSTRACT

**Introduction** People living with and dying from multimorbidity are increasing in number, and ensuring quality care for this population is one of the major challenges facing healthcare providers. People with multimorbidity often have a high burden of palliative and end-of-life care needs, though they do not always access specialist palliative care services. A key reason for this is that they are often not identified as being in the last stages of their life by current healthcare providers and systems. This scoping review aims to identify and present the available evidence on how people with multimorbidity are currently included in research, policy and clinical practice.

**Methods and analysis** Scoping review methodology, based on Arksey and O'Malley's framework, will be undertaken and presented using the Preferred Reporting Items for Systematic Review and Meta-Analyses extension for Scoping Reviews. Search terms have been generated using the key themes of 'multimorbidity', 'end of life' and 'palliative care'. Peer-reviewed research will be obtained through systematic searching of Medline, EMBASE, CINAHL, Scopus and PsycINFO. Grey literature will be searched in a systematic manner. Literature containing a definition for adults with multimorbidity in a terminal phase of their illness experience will be included. After screening studies for eligibility, included studies will be described in terms of setting and characteristics as well as using inductive content analysis to highlight the commonalities in definitions.

**Ethics and dissemination** Ethical approval is not required for this scoping review. The findings of the scoping review will be used internally as part of SPB's PhD thesis at the University of St Andrews through the Multimorbidity Doctoral Training Programme for Health Professionals, which is supported by the Wellcome Trust (223499/Z/21/Z) and published in an open access, peer-reviewed journal for wider dissemination.

## STRENGTHS AND LIMITATIONS OF THIS STUDY

⇒ This is the first scoping review to identify and map out how people with multimorbidity who are approaching the end of their life are identified in research, policy and practice.
⇒ This protocol has been developed robustly using international guidelines and will be reported using the Preferred Reporting Items for Systematic Review and Meta-Analyses extension for Scoping Reviews guidelines.
⇒ Using content analysis will allow for definitions from various bodies of work to be described narratively and aid future researchers, policy makers and clinicians.
⇒ A project patient advisory group has been established who participated in conceptualisation of this scoping review.
⇒ This work is limited to English language but there is no date limit used.

## INTRODUCTION

Multimorbidity, defined as the presence of two or more physical or mental health conditions, is one of the main challenges facing modern healthcare systems.[1] The prevalence of multimorbidity is increasing and is particularly high in older adults, those living with socioeconomic deprivation and those with mental health disorders.[2] People living with multimorbidity are known to have poorer quality of life, increased healthcare utilisation and expenditure and shorter life expectancy.[3–6] The National Institute for Health and Care Excellence (NICE) published guidance on managing multimorbidity with an aim to reduce treatment burden and unplanned care through providing individualised management plans and patient-centred care.[7] While not focusing on how to deliver palliative and end-of-life care, this guidance acknowledged further research is needed to identify people with multimorbidity who have a reduced life expectancy.

People with multiple long-term conditions are inherently dealing with the prospect that their health conditions are not curable and that these may directly, or indirectly, lead to a shortening of their life expectancy.[8–10] For people with incurable single disease conditions, identification of potential unmet palliative and end-of-life care needs is supported by

various guidelines which advocate for the proactive adoption of a palliative care approach and, where needed, early referral to palliative care services.[11 12] Palliative care is defined by the World Health Organisation (WHO) as: 'an approach that improves the quality of life of patients (adults and children) and their families who are facing problems associated with life-threatening illness'.[13] Informal, often family, caregivers play a pivotal role in meeting the palliative care needs of their loved ones.[14] Palliative care can also be provided clinically by both generalists (healthcare professionals working primarily in non-palliative care settings) or specialists within the field of palliative care.[15] Involvement with palliative care specialists, particularly at an early stage, has been shown to have benefits on patient quality of life and in certain circumstances improve survival.[16 17] As per the WHO definition, many people with multiple long-term conditions have been shown to have a variety of palliative care needs, and there is some evidence these are comparable to those with incurable cancer.[18 19] Prediction modelling has also demonstrated that people with multiple long-term conditions are going to be the highest users of palliative care services by 2040.[20] Despite acknowledging the high level of need, people with multimorbidity continue to be under-referred to palliative care services, and/or referred too late.[19 21]

Difficulty in recognising the palliative care needs of people with multimorbidity may arise from challenges in identifying when they are approaching the end of life, and pursuit of disease-modifying treatments may no longer be beneficial.[22] This so-called 'advanced illness' has previously been defined in the literature as 'late-stage chronic illness when one or more conditions become serious enough that general health and function decline and treatments begin to lose their impact—a state that progresses to end of life'.[23] It is recognised that it can be a challenge to identify the end-of-life period for people with multimorbidity due to the unpredictable disease trajectory they often have, which may in part be due to the heterogeneity of multimorbidity itself.[24] Providing the opportunity for identification of an imminence of the end of life would allow patients with multimorbidity and their loved ones the opportunity to express preferences for care and access formalised social, health and bereavement support.[25 26] However, at present, there is no accepted definition or guidance for when in the multimorbidity illness journey a palliative care approach may be beneficial. In other words, the lack of an agreed working definition of 'advanced multimorbidity' risks people with multimorbidity who are nearing the end of life, and their caregivers, missing out on important care and support.

### Aim and objectives
This scoping review aims to identify and present the available evidence on how people with multimorbidity are currently included in research, policy and clinical practice. Specifically, the objectives within this scoping review are to:

► Describe how states of advanced multimorbidity are operationalised within published research and end-of-life care policies.
► Summarise the characteristics and methodologies of the studies examining or including advanced multimorbidity.
► Present the limitations and identify research gaps within this pre-established knowledge.

### METHODS
Scoping review methodology was selected as this particular review type has a role in conceptual mapping in healthcare, particularly in areas without a universally agreed definition.[27] This scoping review will adhere to the five-step Arksey and O'Malley framework for scoping reviews, with reference to both the updates to this from Levac *et al* and the Joanna Briggs Institute (JBI) guidance for conducting scoping reviews.[28–30] This scoping review was reported in accordance with the Preferred Reporting Items for Systematic Review and Meta-Analyses extension for Scoping Reviews (PRISMA-ScR).[31]

### Stage 1: identifying the research question
This review will particularly focus on when the goals of care of those with multimorbidity is moved from solely aiming to reverse or treat chronic conditions, and instead focuses on alleviating suffering and addressing multidimensional needs—in line with a 'palliative care approach'.[13] This is in keeping with standard accepted definitions of other life-limiting conditions and reviews on language around what encompasses an advanced illness—thus the term 'advanced multimorbidity' has been used for this review.[23 32–35] Understanding the definitions of advanced multimorbidity that are already used in research and clinical practice will allow for informed research exploring the needs of this group of patients and how they are identified for both research, clinical care and policy.

### Stage 2: identifying relevant studies
SPB will identify relevant studies through searches of peer-reviewed and grey literature. Search terms relating to multimorbidity, advanced illness and palliative care were developed with expertise from the University of St Andrews library service to build our search strategy (see online supplemental file 1). Patients involved in health research on the impact of multimorbidity, alongside our own public advisors for this work, have denounced the term multimorbidity as having negative terms, with preferences being made for 'multiple long-term conditions', with literature often using the terms interchangeably and therefore we have integrated multiple long-term conditions into our search terms.[36] The databases Medline, EMBASE, CINAHL, Scopus and PsycINFO will be searched for published journal articles.

There will also be a search of grey literature within the UK to identify both unpublished work, indexed relevant

theses and clinical guidelines and policies. Following the guidance set out by Godin *et al*,[37] our grey literature search will comprise a four-step approach of: searching a grey literature database—National Grey Literature Collection, supported by Health Education England[38]; a contained Google search; browsing targeted websites—related to end-of-life care in the UK; and a discussion among the clinical members of the research team (SPB, LM, DW, SEEM and JB) who have experience in generalist and specialist palliative care in various capacities to act as content experts and identify other items for possible inclusion in the review. Any pieces of unpublished work (eg, conference abstracts or preprints) will be followed up with the authors to enquire about subsequent publication. It was deemed important to include only published research in our final texts to ensure that the studies with appropriate definitions had undergone peer-review. Given that scoping review methodology does not typically include assessment of quality, the inclusion of peer-reviewed articles allows for a step to improve the quality of included papers and the review itself.[30 39]

The references of all included reports or articles and their subsequent citations will be searched for additional studies. There will be no date limit on our searches though we will limit to English language.

### Stage 3: study selection

Once all searches are complete, these will be combined and deduplicated using EndNote V.X7. Using the inclusion and exclusion criteria (table 1), developed using the JBI Population-Concept-Context model,[28] titles and abstracts will be screened by two independent reviewers (SPB and either LM or DW). We have particularly chosen to exclude studies on index conditions and comorbidity as although some argue that comorbidity and multimorbidity are interchangeable, the former is disease-focused

**Table 1** Inclusion and exclusion criteria

| Inclusion | Exclusion |
|---|---|
| **Population** | |
| Adults (>18) with multimorbidity (≥2 health conditions) or their carers or healthcare providers | Studies focusing on one index condition<br>Studies focusing on comorbidity |
| **Concept** | |
| Definitions of multimorbidity in a terminal phase of the illness experience | Studies related solely to prognosis<br>No clear definition for advanced multimorbidity in the study/report |
| **Context** | |
| All settings and countries<br>Original research articles and review articles | Unpublished research<br>Not English language |

**Table 2** Data charting

| Study citation information | Author<br>Title<br>Journal<br>Year of publication |
|---|---|
| Study objective | Aim and/or objectives |
| Study methods | Type of study |
| Study characteristics | Country<br>Healthcare setting (if applicable)<br>Participants (number, gender, diagnoses)<br>Inclusion/exclusion criteria |
| Extracted results | Definition for advanced multimorbidity<br>Findings related to aim of study<br>Other relevant information |

whereas the use of multimorbidity allows for delivery of more holistic, patient-centred care.[40] The full texts of selected citations will then be reviewed by two independent reviewers (SPB and either PB, LM or DW) to allow for approval for final study selection. Any discrepancies that may arise from each reviewer will be relayed back to a third reviewer and discussions will be held to reach consensus. Each stage of this study selection will be reported using the PRISMA-ScR flow chart.[31]

### Stage 4: charting the data

Data will be extracted by two independent reviewers to an Excel spreadsheet via the University of St Andrews OneDrive system to allow all researchers access—known as 'data charting' in scoping reviews.[28] Charted data will relate to study design, study population, definition of multimorbidity and reported outcomes, summarised in table 2. This will be an iterative approach with opportunity for updating the data-charting form as informed by the literature at regular research team meetings. Given that scoping reviews aim to assess the breadth of the evidence base, formal quality assessment is not needed.[28]

### Stage 5: collating, summarising and reporting the results

Once we have identified all available definitions in the literature, we will analyse the data, report the results and apply meaning to the results. We will first summarise the characteristics of the included work descriptively by listing the overall number of studies, distribution of the work, study populations and countries. Using content analysis, we will explore the different definitions that are found. Content analysis allows for analysis of both quantitative and qualitative research and policies to describe phenomena to make replicable and valid inferences about the data and provide a practical guide to action. It has been successfully used in other scoping reviews to assimilate data on definitions.[41–43] A qualitative, inductive, content analysis approach was selected as this is preferred when there is limited knowledge about the phenomenon of choice.[44] Given there is no formally accepted standard

definition of advanced multimorbidity, we will undertake our content analysis inductively which involves three main phases:

► Preparation phase: where the unit of analysis (in this case the definition) is selected from the data using keywords and phrases and a description of the context in which this was obtained.

► Organisation phase: open coding is then used to collate broad categories to describe the definitions which are then organised.

► Abstraction phase: this formulates a general description of the research topic through the categories above and will then be described narratively. This may include understanding how definitions vary between different disease groupings, as multimorbidity is a heterogeneous condition with variability in which combinations of conditions it is composed of.[44]

## Patient and public involvement

The concept and aim of this scoping review has been reviewed by our public advisory group, who are members of the Fife Community Advisory Council providing patient and public involvement to the University of St Andrews. The group felt that identifying the dying phase of an individual's life was important, though also emphasised that understanding of the individual at the core of this was of most importance. This led us to use more holistic, patient-centred themes to our search terms (including multiple long-term conditions and the decision to focus on multimorbidity and not comorbidities). The group also felt that uncoordinated care was a challenge for people at the end of life and it is hoped this scoping review will provide operational suggestions of ways to identify people to then provide holistic, co-ordinated services in the future. The public advisory group will contribute to add their perspective to our results, particularly at the abstraction phase to ensure these are meaningful and relevant to people with lived experience.

## Ethical approval and dissemination

This scoping review will not require ethical approval.

The findings of this scoping review will be shared through professional networks, conferences and published open access in a peer-reviewed journal.

## DISCUSSION

This scoping review aims to comprehensively analyse the currently used definitions to identify people with multimorbidity who may soon be approaching the end of life. Identifying patterns to these definitions will allow for understanding which are the important issues to consider when identifying advanced multimorbidity. This will primarily allow for earlier advanced care planning and opportunities for specialist care for patients with multimorbidity as well as ensuring more consistent research is undertaken and impact on national and international policy development.

**Acknowledgements** We are extremely grateful to Vicki Cormie, Senior Librarian at University of St Andrews, for her support and guidance in the development of the search strategy and to our public advisory team who have provided their expert lived opinion to the development of this scoping review. Particular thanks also to Rosie Dunn for her guidance on using content analysis within scoping reviews.

**Contributors** SPB developed the concept of the scoping review, research questions, public advisory group discussions and drafted the manuscript. Input to conceptualisation was also given by SEEM, UA, LW, FQ and JB. SEEM and UA assisted with development of the search strategy. SPB, LM, DW, SEEM and JB provided content guidance through their clinical expertise. SPB drafted the manuscript with PB, LM, DW, FQ and JB critically reviewing drafts of this. All authors approved the final submitted manuscript and agreed to be accountable for all aspects of this protocol.

**Funding** This review will contribute to the primary author's (SPB) PhD Thesis. SPB is a fellow on the Multimorbidity Doctoral Training Programme for Health Professionals, which is supported by the Wellcome Trust (223499/Z/21/Z), and supervised by SEEM, UA, LW, JB and FQ. PB is also a fellow on the Multimorbidity Doctoral Training Programme for Health Professionals, which is supported by the Wellcome Trust (223499/Z/21/Z). As this research was funded in whole, or in part, by the Wellcome Trust (223499/Z/21/Z), for the purpose of open access, the author has applied a CC BY public copyright licence to any author accepted manuscript version arising from this submission.

**Competing interests** None declared.

**Patient and public involvement** Patients and/or the public were involved in the design, or conduct, or reporting, or dissemination plans of this research. Refer to the Methods section for further details.

**Patient consent for publication** Not applicable.

**Provenance and peer review** Not commissioned; externally peer reviewed.

**ORCID iD**
Sarah P Bowers http://orcid.org/0000-0003-0722-8318

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
