## [Reviewer comments · BMJ Open]

ARTICLE DETAILS

TITLE (PROVISIONAL)	Current definitions of advanced multimorbidity: a protocol for a scoping review.
AUTHORS	Bowers, Sarah; Black, Polly; McCheyne, Lewis; Wilson, Darcy; Mills, Sarah; Agrawal, Utkarsh; Williams, Linda; Quirk, Frances; Bowden, Jo

VERSION 1 – REVIEW

REVIEWER	Etkind, Simon University of Cambridge
REVIEW RETURNED	21-Aug-2023

GENERAL COMMENTS	Thanks for the opportunity to review this review protocol. I think the review addresses an important question and the methodology is appropriate, with good use of PPI. I have a few comments and suggestions for improvement. General points 1. The main issue I'd like more clarity on is the aim, which I don't quite understand, and which is described quite differently in the abstract and main text. Is your overarching aim to understand how multimorbidity is defined in the literature/policy? Or are you aiming to develop a new operational definition for advanced multimorbidity? Or both? Recommend you have a single sentence stating the aim which is consistent through the paper. For the aim and objectives section I would then suggest you move some of the detail in the aim paragraph to the objectives list or the methods Specific points Title: 1. Suggest state overtly in the title that this is a protocol paper "protocol for a scoping review" Abstract 2. If possible, it would be helpful to have some info about the inclusion criteria here. You may be able to shave some words off the introduction section of the abstract to fit this in. Introduction: 3. This includes relevant literature to set the scene for the review and I think I follow your argument, but it could be presented more clearly. Suggest minor revisions to focus more along the lines of the following:1. Multimorbidity is common and increasing2. people with multimorbidity in later stages have high palliative care needs
--

	3. whilst there are well understood mechanisms to identify the advanced stages of single illnesses, this is much more challenging in multimorbidity, and poor identification has impacts on care planning etc. 4. A definition of advanced multimorbidity is needed to address point 3 4. last sentence of paragraph 1: I'm not sure how the second phrase in this sentence links in with your wider argument? 5. First sentence of paragraph 2. If you are using multiple long-term conditions as per your PPI group, then the norm will be that these are not curable. So this sentence might need to be rephrased. 6. I'm not sure you need the full definition of palliative care in paragraph 2, it distracts from the flow and could be shortened. Methods 7. Table 1: You have 'unpublished research' as an exclusion criterion, but in the methods state you are conducting a grey literature search to identify unpublished work. I'm wondering what you will do with articles from the grey literature search – will these be excluded? What will you do with conference abstracts or pre-prints? Could you add more detail here to clarify please. As this is a scoping review I am thinking you may want to include quite a broad range of article types? 8. I agree reference list searching will be helpful, and suggest you also include citation list searching of included articles to help identify the latest literature. 9. Recognise you are following accepted practice for scoping reviews, but if you aren't using formal quality assessment, I'm wondering you will decide what weight to give different included articles in analysis. Or is this not relevant for your proposed analysis? 10. Search strategy. I recognise you've had expert input into this, and it looks comprehensive. But I am wondering about step 9. I don't quite understand why this bit is separate to the first two search concepts, how does it differ? Could you add an explanation of why you chose this search structure in the supplementary file 11. Scope of analysis. Will you be exploring variation in definitions of advanced multimorbidity based on different disease groupings? e.g. do definitions vary between studies that include mental health conditions compared to those focusing purely on physical conditions? Or vary with different combinations of conditions? I realise it may be difficult to say whether this is going to be feasible until you have more of a sense of the range of the literature, but might be worth mentioning what you will be looking for in relation to how definitions of multimorbidity vary.
--	--

REVIEWER	Czypionka, Thomas
	Institut fur Hoehere Studien, Health Economics and Health Policy
REVIEW RETURNED	12-Sep-2023

GENERAL COMMENTS	Thank you for inviting me to review this protocol on advanced multi-morbidity.
--

	In the introduction, I find this sentence on palliative care puzzling: 'It can be delivered by both generalists (e.g. general practitioners, secondary care practitioners) or specialists within the field of palliative care.(16)'. I am not sure why the authors leave all formal and informal LTC and other professions out of palliative care. The same applies to this sentence: 'As per the WHO definition, people with multiple long-term conditions have been shown to have a variety of palliative care needs, and there is some evidence these are comparable to those with incurable cancer.(19, 20)'. Not all people with multiple long-term conditions have palliative care needs. The introduction may also benefit from mentioning goal-oriented medicine, as the authors aim at advanced stages of multiple chronic conditions. It is an important concept in this area. Methods section: The following sentence is unclear 'The full texts of selected citations will then be reviewed by TWO independent reviewers (SPB, LM and DW) to allow for approval for final study selection'. Three names are given. Referring to content analysis (e.g. page 9 line 32), authors should be more specific (there are several approaches) and cite a reference, as there are multiple traditions of content analysis.
--	---

VERSION 1 – AUTHOR RESPONSE

Reviewer	Comment	Action taken
General		
1	The main issue I'd like more clarity on is the aim, which I don't quite understand, and which is described quite differently in the abstract and main text. Is your overarching aim to understand how multimorbidity is defined in the literature/policy? Or are you aiming to develop a new operational definition for advanced multimorbidity? Or both? Recommend you have a single sentence stating the aim which is consistent through the paper. For the aim and objectives section I would then suggest you move some of the detail in the aim paragraph to the objectives list or the methods	Thank you for highlighting this important point. From this scoping review, we will highlight some of the commonalities and gaps that exist in the definitions currently used. The intention has not been to develop a new operational definition, though some key commonalities may emerge through the content analysis. The wording of the aim has been made consistent throughout and the aims and objectives section has been simplified. The detail around the conceptualisation of the term advanced multimorbidity has been moved to the "Identifying the Research

		Question” stage 1 of the methods.
Title		
Editor	In the title, please state that your manuscript is a study protocol.	Study title has been changed to reflect this is "a protocol for a scoping review”
1	Suggest state overtly in the title that this is a protocol paper “protocol for a scoping review”	
Abstract		
Editor	Please place the Strengths and Limitations section directly after the abstract.	This section has been moved as advised
1	If possible, it would be helpful to have some info about the inclusion criteria here. You may be able to shave some words off the introduction section of the abstract to fit this in.	The abstract has been amended to include information on inclusion criteria.
Introduction		
1	This includes relevant literature to set the scene for the review and I think I follow your argument, but it could be presented more clearly. Suggest minor revisions to focus more along the lines of the following:  1. Multimorbidity is common and increasing 2. people with multimorbidity in later stages have high palliative care needs 3. whilst there are well understood mechanisms to identify the advanced stages of single illnesses, this is much more challenging in multimorbidity, and poor identification has impacts on care planning etc. 4. A definition of advanced multimorbidity is needed to address point 3 	We have reviewed and revised the introduction to provide better narrative flow and clarity to address these points. We have kept in the additional section around palliative care as feel this is helpful to aid understanding around palliative care and palliative care services. Clarity has been added to the term “advanced multimorbidity” which is introduced in the introduction with further detail added in the methods.
1	last sentence of paragraph 1: I’m not sure how the second phrase in this sentence links in with your wider argument?	This sentence has been reworded to emphasise that there have been calls for further research on how to identify people with multimorbidity who have a reduced life expectancy.

1	First sentence of paragraph 2. If you are using multiple long-term conditions as per your PPI group, then the norm will be that these are not curable. So this sentence might need to be rephrased.	This sentence has been changed to emphasise that by nature of being long-term, these conditions are not curable, and emphasis to our point that these conditions may in turn lead to a shortened life expectancy.
1	I'm not sure you need the full definition of palliative care in paragraph 2, it distracts from the flow and could be shortened.	This has now been shortened
2	I find this sentence on palliative care puzzling: 'It can be delivered by both generalists (e.g. general practitioners, secondary care practitioners) or specialists within the field of palliative care.(16)'. I am not sure why the authors leave all formal and informal LTC and other professions out of palliative care.	Apologies for any misunderstanding created here. The use of the word practitioner was used to mean healthcare professionals. Our point was to distinguish between primary and secondary palliative care. However the inclusion of informal caregivers in the end of life care of patients is extremely valuable. We have amended this section to provide clarity and recognise the societal and professional roles that you have mentioned. Furthermore, we have clarified that although Palliative Care is a specialty, palliative care is also a specific approach to holistic care for people with life-threatening illness.
2	The same applies to this sentence: 'As per the WHO definition, people with multiple long-term conditions have been shown to have a variety of palliative care needs, and there is some evidence these are comparable to those with incurable cancer.(19, 20)'. Not all people with multiple long-term conditions have palliative care needs.	This sentence has been amended to clarify that many but not all people with MLTCs have a variety of palliative care needs.
2	The introduction may also benefit from mentioning goal-oriented medicine, as the authors aim at advanced stages of multiple chronic conditions. It is an important concept in this area.	I acknowledge that there are many different terms which overlap/come within palliative care including 'realistic medicine', 'shared decision making', 'personalised care',

		'goal oriented care' - whilst I haven't explicitly described all such terms, the implication is that they are all part of the palliative care approach.
Methods		
Editor	Please include the Patient and Public Involvement statement as a sub-heading in the methods section.	This section has been moved as advised
1	Table 1: You have 'unpublished research' as an exclusion criterion, but in the methods state you are conducting a grey literature search to identify unpublished work. I'm wondering what you will do with articles from the grey literature search – will these be excluded? What will you do with conference abstracts or pre-prints? Could you add more detail here to clarify please. As this is a scoping review I am thinking you may want to include quite a broad range of article types?	A statement has now been added to our methods (Stage 2: Identifying Relevant Studies) to highlight that we will follow up unpublished research with the authors to enquire about subsequent publication. We have chosen to only include published research to ensure the studies (and therefore the included definitions) have undergone peer-review. Including only peer-reviewed studies also helps ensure included articles are of high quality (as per the point addressing quality assessment below)
1	I agree reference list searching will be helpful, and suggest you also include citation list searching of included articles to help identify the latest literature.	Thank you, citation searching has now also been added to this section.
1	Recognise you are following accepted practice for scoping reviews, but if you aren't using formal quality assessment, I'm wondering you will decide what weight to give different included articles in analysis. Or is this not relevant for your proposed analysis?	As you have acknowledged, quality assessment does not form a routine part of scoping and we will not be weighting the articles in our analysis. For scoping review methodology the inclusion of peer-reviewed articles is seen as a way to ensure studies are of a high quality. Therefore whilst not a good fit for this particular review we have instead given consideration to quality

		through selection of peer reviewed papers only.
1	Search strategy. I recognise you've had expert input into this, and it looks comprehensive. But I am wondering about step 9. I don't quite understand why this bit is separate to the first two search concepts, how does it differ? Could you add an explanation of why you chose this search structure in the supplementary file	Thank you for highlighting this. Step 9 was added in discussion with our research librarian to specifically look at when phrases such as advanced multimorbidity were used. After some initial trial searches, it was found to be a necessary step to ensure the search was comprehensive. The search strategy now has a statement included to describe it.
1	Scope of analysis. Will you be exploring variation in definitions of advanced multimorbidity based on different disease groupings? e.g. do definitions vary between studies that include mental health conditions compared to those focusing purely on physical conditions? Or vary with different combinations of conditions? I realise it may be difficult to say whether this is going to be feasible until you have more of a sense of the range of the literature, but might be worth mentioning what you will be looking for in relation to how definitions of multimorbidity vary.	Thank you for your thought provoking question/suggestion. As part of our data extraction (charting the data) we will extract data including participant diagnoses (e.g. different disease groupings) and the description of the definition of advanced multimorbidity). As part of our analysis we should hopefully be able to describe differences between the included sub-groups of multimorbidity that are included in the full texts. An additional line has been added to our Abstraction phase to clarify this point and to enable a greater level of description in our findings.
2	The following sentence is unclear 'The full texts of selected citations will then be reviewed by TWO independent reviewers (SPB, LM and DW) to allow for approval for final study selection'. Three names are given.	The wording of this section has been changed to clarify that full texts will be reviewed by SPB and either LM or DW.
2	Referring to content analysis (e.g. page 9 line 32), authors should be more specific (there are several approaches) and cite a reference, as there are multiple traditions of content analysis.	Our justification for choosing this content analysis approach has been further explained in Stage 5 of the methods, highlighting the reference we used for this.

		(Elo S, Kyngäs H. The qualitative content analysis process. J Adv Nurs. 2008;62(1):107-15.)
--	--	---

VERSION 2 – REVIEW

REVIEWER	Etkind, Simon University of Cambridge
REVIEW RETURNED	07-Oct-2023

GENERAL COMMENTS	Thanks for making revisions to incorporate my earlier suggestions. The protocol is much clearer now and I'd be happy for it to be published in its current form. This will be an important review and I'm excited to see your findings in due course. Two minor things to consider  1. I'm not sure 'included' is the right word for the aim. Would 'identified' be a better word? 2. I think there is a grammatical error in the sentence in the first part of the methods "This review will particularly focus on when the goals of care of those with multimorbidity is moved from solely aiming to reverse or treat chronic conditions". Shouldn't it be 'goals of care are' or 'goal of care is'
---

REVIEWER	Czypionka, Thomas Institut fur Hohere Studien, Health Economics and Health Policy
REVIEW RETURNED	25-Oct-2023

GENERAL COMMENTS	I am looking forward to the results to of the scoping review!
---